# Flows of Dense Suspensions of Polymer Particles through Oblique Bifurcating Channels: Two Continua Approach

**DOI:** 10.3390/polym14183880

**Published:** 2022-09-17

**Authors:** Vladimir Shelukhin, Andrey Antonov

**Affiliations:** Lavrentyev Institute of Hydrodynamics, 630090 Novosibirsk, Russia

**Keywords:** suspensions, two-velocity continua, bifurcating channel, partitioning of particles, non-Newtonian rheology, pulsatile injection

## Abstract

A two-velocity mathematical model is proposed for dense suspension flows through channel bifurcations. Equations agree with thermodynamic laws and they are suitable for both heavy and light particles. The pulsatile mode of injection of particles is considered. In the 2D-case, we address the issue of partitioning particles and study how a loss of particles into the side branch depends on the bifurcation angle. A qualitative agreement with experiment data are established. We capture the Zweifach–Fung effect. We treat polymer particles as a phase enjoying the rheology of the Bingham viscoplastic material. We prove that the polymer particle distribution between two branches correlates with the averaged-in-time Bingham number in these branches.

## 1. Introduction

Suspension flows through bifurcating channels are relevant to many natural and technological processes. Their application concerns medicine, biotechnology and microfluidic devices. Micromoulding of thermoplastic polymer is a developing process with great potential for producing low-cost microfluidic devices. Among different micromoulding techniques, micro-injection moulding is one of the most promising processes suitable for manufacturing polymeric disposable microfluidic tubes with branches. The review paper [1] presents the main significant developments that have been achieved in different aspects of micro-injection moulding of microfluidic devices. Aspects covered include device design, machine capabilities, mould manufacturing, material selection and process parameters. Problems, challenges and potential areas for research are highlighted. One more application of the present paper concerns the control of proppant flow through perforations in the wellbore. Such a problem is important in hydraulic fracturing technology [2].

We develop a thermodynamically consistent two-velocity model of a granular fluid, taking into account the difference in the rheology of the carrier fluid and the solid phase consisting of particles. There is no assumption on dilute suspensions, and the model allows one to describe such processes as the deposition of particles. The validation of the model was carried out on a benchmark experiment known as the Boycott effect [3], implying that an inclination of a vertical vessel enhances the sedimentation of particles in suspensions. Historically, suspensions have been considered within the framework of a single-phase continuum with a viscosity dependent on particle concentration. Currently, two-phase approaches are being used more and more. They are usually based on methods of averaging either over a volume or over an ensemble of particles. In this paper, when deriving equations, we follow a different method, which was originally developed by Khalatnikov and Landau for the thermodynamics of superfluid helium 2He [4]. According to this method, the forces of interfacial interaction for reversible processes are first uniquely determined, since the energy conservation makes the entire system of conservation laws overdetermined. Then, the dissipative forces of phase interaction are determined by matching the conservation laws with the general principles of irreversible de Groot–Mazur thermodynamics [5].

In a number of works on two-phase flows, additional forces are introduced into the momentum equations, such as the force of Archimedes, Stokes, Suffman, Magnus. Even forces that depend explicitly on time are used (the Basset–Boussinesq force) [6]. In this work, instead of additional forces, we use the generalized Fick diffusion law for the particle mass concentration flux vector. This approach is consistent with thermodynamics and proved to be successful in sedimentation problems if the gravitational component is taken into account in Fick’s law. The generalized Fick’s law takes into account not only the concentration gradient, but also the gradients of temperature, pressure, and the modulus of the phase velocity difference. The Stokes resistance force is taken into account through the coefficient of interfacial friction, which satisfies the known correlations. The diffusion approach we use imposes limitations on the applicability area. The particles must be sufficiently small.

In the present paper, the problem of the flow when there is a branch in the main channel is considered. The pulsatile regime, when the injection of particles alternates with the injection of a clean fluid, is studied in detail. In a number of papers [7], suspensions of polymer particles are treated as non-Newtonian fluids. Here, we follow the same approach and the particulate phase is represented by the rheology of a viscoplastic Bingham fluid. We establish that the loss of particles into the branching channel is considered as the smaller, the greater the angle of branching from the direction of the main channel. The dependence of the flow on the viscosities and the yield stress of the granular phase is addressed.

Recently, interest in the flow of suspensions through channel bifurcations has been growing in microfluidics and biology due to the problem of particle separation [8]. Partitioning of erythrocytes, leukocytes, and platelets at vascular junctions plays an important role in determining microvascular hematocrits and has important physiological implications. One of the advantages of continuum models is that solvers known in computational fluid mechanics can be applied to them. In addition, these solvers can be easily adapted to complex geometries and non-Newtonian fluid rheology.

A number of approaches have been developed, which use external forces to affect the fluid/particle behavior in microchannel-based separation systems. Such forces can be magnetic [9], optical [10], acoustic [11], chemical [12], thermal [13], gravitational [14,15], electrical and electronic [16,17]. The most popular separation method is based on gravitation. However, this technique is not as effective in microchannel devices because of the low Reynolds number, laminar flow and interface tension [18]. In macro-scale separation systems, gravitational approaches are based on the forces of inertia to help speed up the separation process, an example of this being centrifugal sorting [19]. This has led to an interest in adopting new microscopic separation methods, in which viscous forces, shear strain rate, interface tension and the geometrical effects—often used for inertial focusing—play an important role. Various devices have been designed that make use of the effects of hydrodynamics and viscous forces. Examples include microfilter devices [20,21] and microchannel devices [22,23]. Furthermore, the bending channel [24,25], channel constriction [26,27], and bifurcated channels [28,29] have been explored. In the bending channel design, inertial effects are exploited whereby a centrifugal force is produced on the particles when passing through the curved section of the channel and the skimmed liquid is likely to be formed in the central region. Flow velocity is a key factor for the effectiveness of this design. A channel constriction is used to force particles to move to the channel center.

Fluid separation in T-shaped microchannels is studied in many papers, see, for example [30]. Xi and Shapley studied the flow of concentrated suspension in an asymmetric T-junction bifurcation of rectangular channels with nuclear magnetic resonance imaging [31]. They observed that the particles are almost equally partitioned between the downstream branches, and this indicates the migration of particles across the dividing streamlines near the bifurcation section.

In the present paper, we present a modeling analysis for dense suspension flows through oblique bifurcating channels. Such an issue has been addressed in [32] for steady flows and in the case of neutrally buoyant particles, and in the low Reynolds number regime. We focus on the difference between fluid and particle velocities, interphase friction, viscous forces, non-Newtonian rheology and geometrical effects. Particularly, we consider in detail the pulsatile mode of injection of particles and study how the loss of particles into the side branch depends on the bifurcation angle. We prove that the partitioning of particles occurs in agreement with the Zweifach–Fung effect, stating that particles prefer the high-flow-rate branch. Our method is suitable for both heavy and light particles. Qualitative agreements with experiment data are established. Paying emphasis on pulsatile injection of polymer particles, we establish that more particles fall into the branch with a lower mean Bingham number. A numerical algorithm is developed within the framework of the FreeFem++ package. In the incompressible case, a modified SIMPLE method is applied in order to ensure the divergence-free condition for the average volumetric velocity. The stability test was carried out by mesh refinement. The approach has been validated in the previous authors’ paper concerning the Boycott sedimentation effect [33].

## 2. Mathematical Model

We consider a joint flow of two continua when an arbitrary volume *V* contains a fluid (index *f*) and a granular phase (index *s*). Volume, mass and pressure of the fluid and the granular phases is denoted by $V_{f},m_{f},p_{f}$ and $V_{s},m_{s},p_{s}$, respectively. It is assumed that the granular phase is a mixture of dry particles and a carrier fluid, such as proppant gel. In this case, $V_{s}=V_{M}+V_{p}$ and $m_{s}=m_{M}+m_{p}$ where $V_{M}$ is the mud volume, $V_{p}$ is the volume of dry particles, $m_{M}$ is the mud mass, and $m_{p}$ is the mass of dry particles.

The particles are "frozen" in the carrier liquid, i.e., the granular phase is characterized with just one speed $v_{s}$, one viscosity and one stress tensor. In what follows, the indexes *f* and *s* stand for fluid and solid phases, respectively.

We pass to quantities assigned to the unit volume:(1)$$\rho =\frac{m}{V},\;\rho_{s}=\frac{m_{s}}{V},\;\rho_{f}=\frac{m_{f}}{V},\;\rho_{p}=\frac{m_{p}}{V},\;\phi_{j}=\frac{V_{j}}{V},\;\rho_{M}=\frac{m_{M}}{V},\;c=\frac{m_{p}}{m}.$$

Here, $c=\rho_{p}/\rho$ is the particle mass concentration and $\phi_{j}$ is the volume fraction of the *j*-phase with j=f,p,M. It follows from the above definitions that the partial densities $\rho_{j}$ are related to the material densities ${\bar{\rho }}_{j}$ by the following equations
$$\rho_{j}=\phi_{j}{\bar{\rho }}_{j},\;{\bar{\rho }}_{j}\equiv \frac{m_{j}}{V_{j}},\;\phi_{f}+\phi_{s}=1,\;\phi_{s}=\phi_{p}+\phi_{M},\;\rho =\rho_{f}+\rho_{p}+\rho_{M}.$$

Generally, the phase pressures $p_{s}$ and $p_{f}$ are different. However, as in [34], we assume that $p_{s}=p_{f}=p$. Such a hypothesis works well when the surface tension at the boundaries separating the phases are negligible.

Let $v_{i}$, $T_{i}$, l, *k* stand for the velocity, the viscous part of the stress tensor, the particle concentration flux vector and the interphase friction coefficient, respectively.

We introduce the tensor notations. Given two vectors a and b, we define the scalar product a·b= $a_{i}b_{i}$. The tensor product a⊗b is a matrix, such that ${\left(a⊗b\right)}_{ij}=$ $a_{i}b_{j}$. The matrix $A^{*}$ stands for the adjoint matrix of *A*, i.e., ${\left(A^{*}\right)}_{ij}={\left(A\right)}_{ji}.$ The *i*-th component of the vector divA is defined by the formula ${\left(\mathrm{div}\;A\right)}_{i}=\partial A_{ik}/\partial x_{k}$.

If we neglect the rotation of particles and thermal effects, then one can derive from [34] the following mathematical model for six unknowns functions $\rho_{s}$, $\rho_{f}$, *p*, *c*, $v_{s}$, $v_{f}$:(2)$$\frac{\partial (\rho_{s}v_{s})}{\partial t}+\mathrm{div}\;\left(\rho_{s}v_{s}⊗v_{s}\right)=-\frac{\rho_{s}}{\rho }\nabla p-\frac{\rho_{s}\rho_{f}}{2\rho }\nabla u^{2}-k\;u+\mathrm{div}\;T_{s}+\rho_{s}g\;,$$
(3)$$\frac{\partial (\rho_{f}v_{f})}{\partial t}+\mathrm{div}\;\left(\rho_{f}v_{f}⊗v_{f}\right)=-\frac{\rho_{f}}{\rho }\nabla p+\frac{\rho_{s}\rho_{f}}{2\rho }\nabla u^{2}+k\;u+\mathrm{div}\;T_{f}+\rho_{f}g,$$
(4)$$\frac{\partial (\rho c)}{\partial t}+\mathrm{div}\;\left(cj+l\right)=0,$$
(5)$$\rho_{st}+\mathrm{div}\;\left(\rho_{s}v_{s}\right)=0,\;\rho_{ft}+\mathrm{div}\;\left(\rho_{f}v_{f}\right)=0,\;$$
where p=p(ρ) is the prescribed state equation, g is the gravitation vector, and
$${\left(\nabla \;p\right)}_{i}=\frac{\partial p}{\partial x_{i}},\;j=\rho_{s}v_{s}+\rho_{f}v_{f},\;\rho =\rho_{s}+\rho_{f},u=v_{s}-v_{f},\;\mathrm{div}\;l\equiv \frac{\partial l_{i}}{\partial x_{i}},\;u^{2}=u\cdot u.$$

Now, we discuss rheology. Given a velocity field v(x), we introduce the corresponding rate of strain tensor *D* and its deviatoric part $D_{d}$, as follows
$$D=\frac{\nabla v+{\left(\nabla v\right)}^{*}}{2},\;D_{d}=D-\left(\mathrm{div}\;v\right)I,\;I_{ij}=\delta_{j}^{i},\;{\left(\nabla v\right)}_{ij}=\frac{\partial v_{i}}{\partial x_{j}},\;{\left|D\right|}^{2}=D_{ij}D_{ij}.$$

The fluid phase is considered to be a viscous Newtonian fluid. It implies that
(6)$$T_{f}=2\eta_{f}D_{d}^{f},$$
with the constant $\eta_{f}$ staying for shear viscosity. As for the compression viscosity, it is assumed negligible. The solid phase contains particles, with *c* being its mass concentration. Given a vanishing number δ and the volume fraction of the solid phase $\phi_{s}$, we define rheology of the solid phase by the regularized Bingham equation:(7)$$T_{s}\left(\phi_{s}\right)=\left[2\eta_{s}\left(\phi_{s}\right)+\frac{\tau (\phi_{s})}{|D_{d}^{s}{|}_{\delta }}\right]D_{d}^{s}{,\;|D|}_{\delta }^{2}={\left|D\right|}^{2}+\delta^{2},\;{\left|D\right|}^{2}=D:D,\;A:B=A_{j}^{i}B_{j}^{i}.$$

Here,
(8)$$\eta_{s}\left(\phi_{s}\right)=\eta_{s}^{0}{\left(1-\frac{\phi_{s}}{\phi_{s}^{*}}\right)}^{-2.5\phi_{s}^{*}}$$
is the viscosity given by the Krieger–Douhgerty empirical closure [35], with $\phi_{s}^{*}$ and $\eta_{s}^{0}$ being the maximal reference value of $\phi_{s}$ and the consistency, respectively. As for for the yield stress, we take it by the correlation formula
(9)$$\tau \left(\phi_{s}\right)=\tau_{0}\cdot \sqrt{\left(1-\phi_{s}\right){\left(1-\frac{\phi_{s}}{\phi_{s}^{*}}\right)}^{-2.5\phi_{s}^{*}}}$$
proposed in [36]. A mathematical proof is provided in [37] to ensure that Equation (Equation 7) is a good approximation for the Bingham fluid.

One more rheological equation is the Fick law [33]:(10)$$l=-\left(\gamma_{2}\nabla c+\gamma_{1}\nabla p+\gamma_{3}\nabla \;u^{2}\right)+\rho cBg.$$

Due to the mass conservation laws (Equation 5), Equations (Equation 2) and (Equation 3) reduce to
(11)$$\rho_{s}\left(\frac{\partial v_{s}}{\partial t}+v_{s}\cdot \nabla v_{s}\right)=-\frac{\rho_{s}}{\rho }\nabla p-\frac{\rho_{s}\rho_{f}}{2\rho }\nabla u^{2}-k\;u+\mathrm{div}\;T_{s}+\rho_{s}g\;,$$
(12)$$\rho_{f}\left(\frac{\partial v_{f}}{\partial t}+v_{f}\cdot \nabla v_{f}\right)=-\frac{\rho_{f}}{\rho }\nabla p+\frac{\rho_{s}\rho_{f}}{2\rho }\nabla u^{2}+k\;u+\mathrm{div}\;T_{f}+\rho_{f}g.$$

Such equations are of use in calculations performed below.

Let us formulate a hypothesis of incompressibility. We assume that the mud volume fraction $\phi_{M}$ is negligible and the densities of materials ${\bar{\rho }}_{f}$, ${\bar{\rho }}_{p}$ and ${\bar{\rho }}_{M}$ are constants. Then, it follows from (Equation 1) that
(13)$$\rho_{s}\approx c\rho ,\;\rho_{f}\approx \left(1-c\right)\rho ,\;\phi_{s}=\frac{c}{R_{0}+c\left(1-R_{0}\right)},\;\phi_{f}=\frac{R_{0}\left(1-c\right)}{R_{0}+c\left(1-R_{0}\right)},$$
where $R_{0}={\bar{\rho }}_{s}/{\bar{\rho }}_{f}$. Observe that the total density ρ and the partial densities $\rho_{j}$ are not constant in contrast to the densities of materials. By the incompressibility assumption, one can derive easily the following formulas:$$\frac{\rho_{s}}{{\bar{\rho }}_{f}}=c\left[1+\left(R_{0}-1\right)\phi_{s}\left(c\right)\right]\equiv r_{s}\left(c\right),\;\frac{\rho_{f}}{{\bar{\rho }}_{f}}=1-\phi_{s}\left(c\right)\equiv r_{f}\left(c\right),\;\rho =\frac{{\bar{\rho }}_{s}}{R_{0}+c\left(1-R_{0}\right)}.$$

The functions $r_{s}\left(c\right)$ and $r_{f}\left(c\right)$ are dimensionless partial densities.

One more consequence of the incompressibility assumption is that the volumetric mean velocity is divergence-free:(14)$$\mathrm{div}\;v=0,\;v\equiv \phi_{s}\left(c\right)v_{s}+\phi_{f}\left(c\right)v_{f}.$$

Equation (Equation 4) is equivalent to
(15)$$\frac{\partial c}{\partial t}+\tilde{v}\cdot \nabla \;c+\rho^{-1}\left(c\right)\mathrm{div}\;l=0,\;\tilde{v}\equiv cv_{s}+\left(1-c\right)v_{f},$$
where $\tilde{v}$ is the mass mean velocity. Thus, we derived a mathematical model for four unknown functions *p*, *c*, $v_{s}$ $v_{f}$, obeying the Equations (Equation 11), (Equation 12), (Equation 14) and (Equation 15). The parameters $\eta_{s}$, $\eta_{f}$, *k*, $\gamma_{j}$ are assumed to be know functions of the mass concentration *c*.

Under the incompressibility hypothesis, pressure is no longer a thermodynamic parameter and does not satisfy the equation of state. It is now included in the list of unknown functions as in the case of Navier–Stokes models of a viscous incompressible fluid. Densities can be restored from equalities (Equation 13).

The diffusion coefficients $\gamma_{j}$ vanish when any phase disappears. As for the friction, we use the correlation formula
(16)$$k\left(c\right)=\frac{3}{4}C_{D}\frac{c{\bar{\rho }}_{f}\left|u\right|}{d_{p}},$$
proposed in [38] where $d_{p}$ is the particle diameter and $C_{D}$ is the particle/fluid friction:$$C_{D}=\left\{\begin{array}{cc} \frac{24}{Re_{p}}\left(1+0.15Re_{p}^{0.678}\right) & if\;Re_{p}<1000,\\ 0.44 & if\;Re_{p}>1000, \end{array}\right.Re_{p}=\frac{d_{p}{\bar{\rho }}_{f}\left|u\right|}{\eta_{f}}.$$

Now, we apply the described governing equations for 2D flows of suspensions in an oblique bifurcating channel (Figure 1) representing a branching system, in which the parent branch divides into two daughter branches (one branch follows the inlet, termed as the main branch; and another bifurcate follows at an angle α with the main branch, termed as the side branch). This type of bifurcating channel is often encountered in the industry, nature, and human body, and one of the important tasks is to find out the bulk suspension and particle partitioning in the daughter branches for a better understanding of the flow behavior.

On the (x,y) plane, the parent branch is directed along the *x*-axes, Figure 1. The side branch is at the angle α relative to the main branch. Let *H*, *V*, $l_{0}$, $t_{0}$ and $\bar{p}$ stand for the reference values of the channel width, suspension velocity, particle concentration flux, time and pressure. We pass to dimensionless variables
$$x^{\prime }=\frac{x}{H},\;y^{\prime }=\frac{y}{H},\;v^{\prime }=\frac{v}{V},\;p^{\prime }=\frac{p}{\bar{p}},\;l^{\prime }=\frac{l}{l_{0}},\;t^{\prime }=\frac{t}{t_{0}},$$
and choose the reference values in such a way that
$$t_{0}=\frac{H}{V},\;l_{0}={\bar{\rho }}_{f}V,\;\bar{p}={\bar{\rho }}_{f}gH.$$

Flows are defined by the following demensionless numbers:Re=HVρ¯fηf,k1=18ηfHρ¯fVdp2,Fr=gHV2,β=k1·Re=18H2dp2,τ˜*=τ0HηfV,
$$\Gamma_{1}\left(c\right)=\frac{\gamma_{1}{\bar{\rho }}_{f}V^{2}}{Hl_{0}},\;\Gamma_{2}\left(c\right)=\frac{\gamma_{2}}{Hl_{0}},\;\Gamma_{3}\left(c\right)=\frac{\gamma_{3}V^{2}}{Hl_{0}},\;\Gamma_{4}\left(c\right)=\frac{Bg}{{\bar{\rho }}_{f}V}c\left[r_{s}\left(c\right)+r_{f}\left(c\right)\right].$$

Dimensionless stress tensors appear:$$T_{f}=\frac{\eta_{f}V}{H}T_{f}^{\prime },\;T_{s}=\frac{\eta_{f}V}{H}T_{s}^{\prime },$$
with
(17)$$T_{f}^{\prime }=2{\left(D^{\prime }\right)}_{d}^{f},\;T_{s}^{\prime }=\left(\frac{2\eta_{s}^{0}}{\eta_{f}}{\left(1-\frac{\phi_{s}}{\phi_{s}^{*}}\right)}^{-2.5\phi_{s}^{*}}+\frac{{\tilde{\tau }}_{*}\sqrt{\left(1-\phi_{s}\right){\left(1-\frac{\phi_{s}}{\phi_{s}^{*}}\right)}^{-2.5\phi_{s}^{*}}}}{|{\left(D^{\prime }\right)}_{d}^{s}{|}_{\delta }}\right){\left(D^{\prime }\right)}_{d}^{s}.$$

In the absence of any phase, the parameters $\gamma_{1}$,$\gamma_{2}$, $\gamma_{3}$ disappear. This is why we impose the formulas
(18)$$\Gamma_{i}=\Gamma_{i}^{0}c\left(1-c\right),\;\Gamma_{4}=\Gamma_{4}^{0}c,\;i=1,2,3.$$

We introduce the mean mass velocity $\tilde{v}=cv_{s}+\left(1-c\right)v_{f}$. Let the differential operators
$$\frac{d_{s}}{dt}=\frac{\partial }{\partial t}+v_{s}\cdot \nabla ,\;\frac{d_{f}}{dt}=\frac{\partial }{\partial t}+v_{f}\cdot \nabla ,\;\frac{d}{dt}=\frac{\partial }{\partial t}+\tilde{v}\cdot \nabla ,$$
stand for the material derivatives.

When omitting primes, we find that the functions $v_{f}^{\prime }\left(x^{\prime },y^{\prime },t^{\prime }\right)$, $v_{s}^{\prime }\left(x^{\prime },y^{\prime },t^{\prime }\right)$, $c(x^{\prime },y^{\prime },t^{\prime })$, $p^{\prime }\left(x^{\prime },y^{\prime },t^{\prime }\right)$ obey the equations
(19)rs(c)Redsvsdt=−cRe∇p−Rers(c)(1−c)2∇u2−βcu+divTs+rs(c)ReFr·eg,
(20)rf(c)Redfvfdt=−(1−c)Re∇p+Rerf(c)c2∇u2+βcu+divTf+rf(c)ReFr·eg,
(21)$$\mathrm{div}v\equiv \mathrm{div}\left[\phi_{s}\left(c\right)v_{s}+\phi_{f}\left(c\right)v_{f}\right]=0,$$
(22)$$\frac{dc}{dt}+\frac{\mathrm{div}l}{r_{f}\left(c\right)+r_{s}\left(c\right)}=0,$$
where
(23)$$l=-\Gamma_{2}\left(c\right)\nabla c-\Gamma_{1}\left(c\right)\nabla p-\Gamma_{3}\left(c\right)\nabla u^{2}+\Gamma_{4}\left(c\right)e_{g},$$

In what follows, we neglect the gravitation effect since we restrict ourselves to 2D horizontal flows.

The flow domain Ω is depicted in Figure 1. Let n stand for the unit outward normal vector to ∂Ω. The boundary conditions are formulated as follows. At the inlet boundary, we set
(24)$$v_{f}=v_{f}^{inlet}\left(t,y\right),\;v_{s}=v_{s}^{inlet}\left(t,y\right),\;c=c^{inlet}\left(t,y\right).$$

The outlet boundary conditions are given as follows:(25)$$\frac{\partial v_{f}}{\partial n}=0,\;\frac{\partial v_{s}}{\partial n}=0,\;\frac{\partial c}{\partial n}=0,\;p=p^{outlet}=\mathrm{const}.$$

At the impenetrable boundaries, we set
(26)$$v_{f}=0,\;v_{s}=0,\;\frac{\partial c}{\partial n}=0.$$

The initial conditions are
(27)$$v_{f}=v_{f}^{0}\left(x,y\right),\;v_{s}=v_{s}^{0}\left(x,y\right),\;c=c^{0}\left(x,y\right).$$

The functions standing in the right-hand sides of Equations (Equation 25)–(Equation 27) are assumed known.

## 3. Numerical Algorithm

Let us write the system (Equation 19)–(Equation 22) in the weak form using the Sobolev space $W_{2}^{1}\left(\Omega \right)$. Given test functions $w_{1},w_{2}\in W_{2}^{1}{\left(\Omega \right)}^{2}$ and $\psi_{1},\psi_{2}\in W_{2}^{1}\left(\Omega \right)$, we apply the boundary conditions (Equation 24)–(Equation 26) to derive the equations
(28)∫∫Ωw1·rs(c)dsvsdt+βcReudxdy++∫∫Ω1Re∇w1:Tsdxdy−∫∫Ωdivw1cp+rs(c)(1−c)2u2dxdy=0,
(29)∫∫Ωw2·rf(c)dfvfdt−βcReudxdy++∫∫Ω1Re∇w2:Tfdxdy−∫∫Ωdivw2(1−c)p−rf(c)c2u2dxdy=0,
(30)$$\begin{array}{c} \underset{\Omega }{\;\int \int \;}\nabla \psi_{1}\cdot \left[\phi_{s}\left(c\right)v_{s}+\phi_{f}\left(c\right)v_{f}\right]\;dx\;dy=0, \end{array}$$
(31)$$\begin{array}{c} \underset{\Omega }{\;\int \int \;}\psi_{2}\frac{dc}{dt}\;dx\;dy-\underset{\Omega }{\;\int \int \;}\frac{\nabla \psi_{2}\cdot l}{r_{s}\left(c\right)+r_{f}\left(c\right)}\;dx\;dy=0, \end{array}$$

We solve the system (Equation 28)–(Equation 31), applying the finite element method embedded into the open source package FreeFem++.

While passing from the time level $t_{m}$ to the level $t_{m+1}$ with $\tau =t_{m+1}-t_{m}=$ const, all the functions
$$v_{f}^{m}=v_{f}\left(x,t_{m}\right),\;v_{s}^{m}=v_{s}\left(x,t_{m}\right),\;u^{m}=v_{s}^{m}-v_{f}^{m},$$
$$p^{m}=p\left(x,t_{m}\right),\;c^{m}=c\left(x,t_{m}\right),\;x=\left(x,y\right)$$
are assumed known.

Over triangulation of the domain Ω, we use piecewise P2-elements for velocities, whereas for concentration and pressure, we use P1-elements.

To calculate the material derivative of a function f(x,t), we use the following approximation:$$\frac{df}{dt}=\frac{\partial f}{\partial t}+v\cdot \nabla f\approx \frac{f\left(x,t_{m+1}\right)-f\left(X^{m}\left(x\right),t_{m}\right)}{\tau },$$
where $X^{m}=X\left(t_{m}\right)$ with X(t) being the solution of the Cauchy problem
$$\frac{dX(t)}{dt}=v^{m}\left(X\left(t\right)\right),\;X\left(t_{m+1}\right)=x,\;t_{m}<t<t_{m+1}.$$

Within FreeFem++, there is a procedure called “convect” which allows to solve this Cauchy problem and to determine the value $f(X^{m}\left(x\right),t_{m})$:(32)$$\begin{array}{c} f\left(X^{m}\left(x\right),t_{m}\right)=convect\left(v^{m}\left(x\right),-\tau ,f^{m}\left(x\right)\right) \end{array}$$

The material derivatives $d_{f}/dt$ and $d_{s}/dt$ are approximated in a similar way.

Let us describe an algorithm.

*Step 1.* To tackle nonlinearity, we apply iterations as far as the concentration *c* is concerned. Given $c^{m}$ at the time level $t_{m}$, we find $c^{m+1}$ at the time level $t_{m+1}$ by iterations. Given function $c^{iter}$, we define the next iteration $c^{iter+1}$ as a solution to the problem
(33)$$\begin{array}{c} \underset{\Omega }{\;\int \int \;}\psi_{2}\frac{c^{iter+1}\left(x,t_{m+1}\right)-convect\left(v^{m}\left(x\right),-\tau ,c^{m}\right)}{\tau }\;dx\;dy\;-\\ -\underset{\Omega }{\;\int \int \;}\frac{\left(\nabla \psi_{2}\cdot l\right)}{r_{f}\left(c^{iter}\right)+r_{s}\left(c^{iter}\right)}\;dx\;dy=0,\\ l=-\left(\Gamma_{1}\left(c^{iter}\right)\nabla c^{iter+1}+\Gamma_{2}\left(c^{iter}\right)\nabla p^{m}+\Gamma_{3}\left(c^{iter}\right)\nabla {\left(u^{m}\right)}^{2}\right)+\Gamma_{4}\left(c^{iter}\right)e_{g}, \end{array}$$

We take $c^{m}$ as the first iteration.

*Step 2.* We calculate the relative discrepancy
(34)$$E_{c}=\frac{\sqrt{\underset{\Omega }{\;\int \int \;}{\left(c^{iter+1}-c^{iter}\right)}^{2}\;dx\;dy}}{\sqrt{\underset{\Omega }{\;\int \int \;}{\left(c^{iter+1}\right)}^{2}\;dx\;dy}}.$$

Steps 1 and 2 are repeated until the condition $E_{c}<0.01$ is met. With the last iteration $c^{final}$ at hands, we define concentration at the level $t_{m+1}$ by the equality $c^{m+1}=c^{final}$.

Next, we solve the hydrodynamic part of equations. To this end, we use a modified SIMPLE method which is one of predictor–corrector approaches. At the prediction step, the guess velocities $v_{s}^{*}$ and $v_{f}^{*}$ are calculated, starting from the current pressure field $p^{S}$. As an initial value for the field $p^{S}$, we use $p^{m}$. Then, a correction *q* to the current pressure is calculated with the help of the guess velocities. The final correction step consists of improving the guess velocities and the current pressure. The process is iterated until it converges, i.e., until the correction to pressure becomes small enough. There is a nonlinearity related to the convective terms in the momentum Equations (Equation 28) and (Equation 29). To tackle such nonlinearity, we use the simple iteration method.

*Step 3.* With the function $c^{m+1}$, $p^{m}$, $v_{f}^{m}$ and $v_{s}^{m}$ being known, we look for $p^{m+1}$, $v_{f}^{m+1}$ and $v_{s}^{m+1}$ by iterations. To this end, we construct a finite sequence of guess pressures $p^{S}$ (S=1,2,⋯) starting from $p^{S}=p^{m}$ when S=1. We associate with $p^{S}$ finite sequences of current velocities $v_{f}^{iter}$ and $v_{s}^{iter}$ (iter=1,2,⋯), such that
$$v_{f}^{iter}=v_{f}^{m},\;v_{s}^{iter}=v_{s}^{m}\;\mathrm{when}\;iter=1.$$

Given $p^{S},\;v_{f}^{iter},\;v_{s}^{iter},$ we define $v_{f}^{*}$ and $v_{s}^{*}$ by solving the following system of equations:(35)∫∫Ωw1rs(cm+1)dsvs*dt+βcm+1Reu*dxdy++∫∫Ω1Re(∇w2:Ts*)dxdy−−∫∫Ωdivw2cm+1pS+rs(cm+1)(1−cm+1)2u*·uiterdxdy=0,
(36)∫∫Ωw2rf(cm+1)dfvf*dt−βcm+1Reu*dxdy++∫∫Ω1Re(∇w2:Tf*)dxdy−−∫∫Ωdivw2(1−cm+1)pS−rf(cm+1)cm+12u*·uiterdxdy=0,

*Step 4.* Given $v_{f}^{*}$ and $v_{s}^{*}$, we find $q^{iter}$ as a solution of the equation
(37)$$\begin{array}{c} \underset{\Omega }{\;\int \int \;}\nabla \psi_{1}\left[\phi_{s}\left(c^{m+1}\right)v_{s}^{*}+\phi_{f}\left(c^{m+1}\right)v_{f}^{*}\right]\;dx\;dy\;-\\ -\underset{\Omega }{\;\int \int \;}\nabla \psi_{1}\left[\frac{\tau \nabla q^{iter}}{r_{f}\left(c^{m+1}\right)+r_{s}\left(c^{m+1}\right)}\right]\;dx\;dy+\\ +\int_{AB}\psi_{1}\left[\phi_{s}\left(c^{m+1}\right)v_{s}^{inlet}+\phi_{f}\left(c^{m+1}\right)v_{f}^{inlet}\right]\cdot \;ds=0, \end{array}$$

*Step 5.* Now, we correct the current velocities, as follows:(38)$$v_{f}^{iter+1}=v_{f}^{*}-\frac{(1-c^{m+1})\tau }{r_{f}\left(c^{m+1}\right)}\nabla q^{iter},\;v_{s}^{iter+1}=v_{s}^{*}-\frac{c^{m+1}\tau }{r_{s}\left(c^{m+1}\right)}\nabla q^{iter}$$

*Step 6.* We calculate the relative discrepancy of velocities
(39)$$E_{v}=\frac{\sqrt{\underset{\Omega }{\;\int \int \;}\left[{\left(v_{f}^{iter+1}-v_{f}^{iter}\right)}^{2}+{\left(v_{s}^{iter+1}-v_{s}^{iter}\right)}^{2}\right]\;dx\;dy},}{\sqrt{\underset{\Omega }{\;\int \int \;}\left[{\left(v_{f}^{iter+1}\right)}^{2}+{\left(v_{s}^{iter+1}\right)}^{2}\right]\;dx\;dy}.}$$

If $E_{v}\geq 0.05$, we substitute $v_{f}^{iter}$ and $v_{s}^{iter}$ in (Equation 35) and (Equation 36) by $v_{f}^{iter+1}$ and $v_{s}^{iter+1}$ and repeat the procedure until the inequality $E_{v}<0.05$ is met. Then, iterations are recognized as convergent and we use the final iterations to define $v_{s}^{SIMPLE}=v_{f}^{final}$ and $v_{s}^{SIMPLE}=v_{s}^{final}$.

*Step 7.* We calculate the relative discrepancy of pressure
(40)$$E_{p}=\sqrt{\frac{\underset{\Omega }{\;\int \int \;}{\left(q^{final}\right)}^{2}\;dx\;dy}{S}},\;S=\underset{\Omega }{\;\int \int \;}1\;dx\;dy.$$

If $E_{p}\geq 10^{-3}$, we substitute $p^{S}$ in (Equation 35) and (Equation 36) by $p^{S+1}=p^{S}+q^{final}$ and repeat steps 3–7 until the inequality $E_{p}<10^{-3}$ is satisfied.

For the time level $t_{m+1}$, we define
$$v_{s}^{m+1}=v_{s}^{SIMPLE},\;v_{f}^{m+1}=v_{f}^{SIMPLE},\;p^{m+1}=p^{m}+q^{final},$$
provided that the condition $E_{p}<10^{-3}$ is satisfied.

## 4. Results

Our principle goal is to estimate the loss of particles into the side branch. We conclude that the greater the bifurcation angle, the smaller the particle loss. To justify this conclusion, we study the pulsatile mode of particles injection when the particle influx alternates with pure fluid influx. A pulsating injection finds many applications in the production of various materials [39].

We denote the total length of the channel AC (Figure 1) by *l*. We assume that the side branch has the length l/2 and bifurcation of the horizontal parent branch is localized at point *L*, such that |BL|=l/2. Figure 2 depicts calculated snapshots of mass concentration *c* for the pulsating mode of particles injection, which corresponds to the following boundary and initial conditions:(41)$$c^{inlet}\left(t\right)=\left\{\begin{array}{cc} 0.95 & t<t_{1},\\ 0.05 & t>t_{2},\\ \mathrm{linear}\;\mathrm{interpolation} & t_{1}<t<t_{2}, \end{array}\right.$$
(42)$$c^{0}\left(x\right)=\left\{\begin{array}{cc} 0.95 & 0<x<x_{1},\\ 0.05 & x>x_{2},\\ \mathrm{linear}\;\mathrm{interpolation} & x_{1}<x<x_{2}, \end{array}\right.$$
where $t_{1}=50$, $t_{2}=60$, $x_{1}=0.01\cdot l$, $x_{2}=0.02\cdot l$ and the particle injection starts at t=0 and stops at t=60. Exactly at t=60, the front of the particles reaches the location of bifurcation. In fact, the pulsating injection given by (Equation 41) and (Equation 42) is a regularization of the ideal pulsating injection, with the numbers 0.95 and 0.05 substituted with 1 and 0, respectively. Such a regularization facilitates calculations.

Figure 2 corresponds to data chosen as follows. We assume that all the branches have the same width *W*. We set
(43)α=30∘,Γ10=10−3,Γ20=10−6,Γ30=Γ40=0,poutlet=10,Re=10,β=105,ρ¯sρ¯f=2,l=16,W=1,τ˜*=1,ηs0ηf=1.

To meet an agreement with no slip boundary conditions at rigid boundaries, we require that the inlet velocities $v_{f}^{inlet}\left(t,y\right)$ and $v_{s}^{inlet}\left(t,y\right)$ have the Poiseuille-like profile versus the vertical variable *y* [40].

Let us discuss findings and implications concerning the results in Figure 2. First of all, the partitioning of particles occurs at the bifurcation of the channel and one can observe the propagation of the concentration wave both in the side and main branches. We emphasize that the particle mass concentration *c* obeys the inequalities 0<c<1 in spite of the fact that it is governed not by the common transport equation but satisfies a complicated diffusion equation with a matrix diffusion coefficient. Observe that we do not use any numerical cutting tricks to ensure the inclusion 0<c<1. The concentration front is not blunted and there is a cusp (inverted V-shape) in the middle in agreement with experimental data [41]. For comparison, we note that the calculations in [32] do not reveal a cusp in the concentration front. The following explanation for the front with cusp can be given. The particle concentration at the center is higher compared to the near-wall regions because the particles migrate from the region of a high shear rate (wall) to the region of a low shear rate (center). Within the framework of our model, estimates show that the transverse phase velocities are very small and the lateral migration of particles is mainly due to diffusion. Since the role of diffusion is significant, there are limitations on the scope. The results are valid for sufficiently small particles.

Let us introduce particle mass fluxes through the cross-sections KL (inlet), MN (outlet) and LN (branch):(44)$$Q_{in}^{p}=\int_{KL}\left(cj+l\right)\cdot e_{x}\;ds,\;Q_{out}^{p}=\int_{MN}\left(cj+l\right)\cdot e_{x}\;ds,\;Q_{br}^{p}=\int_{LN}\left(cj+l\right)\cdot e_{y}\;ds,$$
where the dimensionless total mass velocity j is equal to $r_{f}v_{f}+r_{s}v_{s}$. We observe that such mass fluxes include both a convective and diffusive particle discharge. By choosing the data (Equation 43), we performed a calculation of the introduced fluxes on a pulsation time interval (0,T) great enough to ensure that $Q_{in}^{p}=0$ at t=T.

Integration of the fluxes (Equation 44) over the interval 0<t<T results in the total dimensionless values of the mass of particles:(45)$$M_{in}^{p}=\int_{0}^{T}Q_{in}^{p}\;dt,\;M_{out}^{p}=\int_{0}^{T}Q_{out}^{p}\;dt,\;M_{br}^{p}=\int_{0}^{T}Q_{br}^{p}\;dt.$$

Partitioning of particles between the branches is shown in Figure 3 for $Q_{in}^{p}\left(t\right)$, $Q_{out}^{p}\left(t\right)$, $Q_{br}^{p}\left(t\right)$ and in Figure 4 for $M_{in}^{p}$, $M_{out}^{p}$, $M_{br}^{p}$. One can see that particle mass loss into the side branch occurs. Such a result explains that particle separation in branched channels can really happen due to hydrodynamic forces only. The front of particles reaches the channel-branching location at a moment close to t=100. This is why the mass flow peaks at the cross sections of branches near the bifurcation location occur on the interval 100<t<120. The mathematical model developed here enables one to optimize the separation effect by the variation of geometric, fluidic and other data. Figure 4 is a result of integration of the functions at Figure 3 over the time interval 0<t<T, T=400. Figure 5 shows dynamics of the mean mass concentration of particles through the inlet (KL), outlet (MN) and branch (EF) cross-sections. Almost equal fluxes for the great times into the daughter branches bifurcating at the angle $30^{\circ }$ are due to equality of their widths.

Figure 2 shows the qualitative partitioning of particles at the branching point of the channel with the bifurcation angle at 30 degrees, while Figure 3, Figure 4 and Figure 5 depict the quantitative partitioning. Although the side and main branches have the same width, the flow of particles into the side branch is slightly less. Apparently, this is the effect of inertia. In most papers, attempts to explain the separation of particles are associated with the study of the behavior of streamlines at the bifurcation zone [42]. Our approach is different since we consider unsteady flows. Moreover, we take into account the difference in the velocities of solid particles and fluid. In such a case, the notion of streamline is not applicable.

Figure 6 proves that an increase in the bifurcation angle results in a decrease in the loss of particles into the side branch. A similar conclusion is derived by simulation in [32]. We obtained such a result by performing calculations of the relative mass loss $M_{br}/M_{out}$ for α, taking the values $30^{\circ },45^{\circ },60^{\circ },75^{\circ },90^{\circ },105^{\circ }.$

To compare our numerical results with experiments, we introduce the relative fluid and particles values of mass:(46)$$f=\frac{M_{out}^{f}}{M_{in}^{f}},\;p=\frac{M_{out}^{p}}{M_{in}^{p}},$$
where
(47)$$M_{in}^{f}=\int_{0}^{T}\int_{KL}r_{f}v_{f}\cdot e_{x}ds\;dt,\;M_{out}^{f}=\int_{0}^{T}\int_{MN}r_{f}v_{f}\cdot e_{x}\;dsdt,\;M_{br}^{f}=\int_{0}^{T}\int_{KL}r_{f}v_{f}\cdot e_{y}\;dsdt.$$

Setting $\alpha =90^{\circ }$ and varying the inlet velocities $v_{f}^{inlet}$ and $v_{s}^{inlet}$ in the pulsatile injection regime (Equation 41) and (Equation 42), we conclude that there is a correlation between the relative values of mass *f* and *p* given at Figure 7. One can observe that an increase in *f* causes the value of *p* to increase; this is also in agreement with the available experiment data [43,44]. Although a quantitative comparison with these experiments is not possible due to the 2D assumption of the present simulations, the results in Figure 7 fairly reproduce the experimental trends. We also should note that these experimental results concern very dilute suspensions with c≃0.02.

We verified that inequality
(48)$$\min_{\alpha \in \alpha_{j}}\frac{M_{br}^{f}-M_{out}^{f}}{M_{br}^{p}-M_{out}^{p}}>0.8,\;\alpha_{j}=\left[30^{\circ },45^{\circ },60^{\circ },75^{\circ },90^{\circ },105^{\circ }\right],$$
holds for the data used in this paper. Such a result agrees with the Zweifach–Fung effect [45], amounting to the fact that more particles fall into the branch where more fluid enters. As far as the Zweifach–Fung effect is concerned, we would prefer not only to validate our model, but to address some questions which were not captured by simulations in other publications [45]. First of all, we are interested in the pulsatile mode of injection of particles, while other authors study stationary regimes. The next question concerns particle density. Our method is suitable for both heavy and light particles. All known calculations of flows in the branching channels concern neutrally buoyant particles. We fill this gap.

Within our model, the principle polymer parameters are the viscosity and the yield stress of the solid phase since we treat this phase as a viscoplastic Bingham fluid. Calculations for $\alpha =90^{\circ }$ reveal that pressure can be assumed constant along any cross-section of every branch. Hence, the pressure gradient can be measured along the center line of the branch, |∇p|≃|∂p/∂s|, with *s* standing for the branch-length reckoning from the bifurcation point (*L* for the side branch and *N* for the main branch on Figure 1). On the other hand, it follows from Figure 8 that pressure decreases in *s* at any time instant. Let $|\nabla_{br}\;p|$ and $|\nabla_{out}\;p|$ stand for the pressure drop along the side and main branches:(49)$$|\nabla_{br}\;p|=\frac{{p|}_{FE}{-p|}_{LN}}{l_{br}},\;\left|\nabla_{out}\;p\right|=\frac{{p|}_{NM}{-p|}_{DL}}{l_{out}},$$
where $l_{br}=LF$ and $l_{out}=\left|ND\right|$ are the branch lengths. One can see that $l_{out}<l_{br}$ since |LN|+|ND|= l/2 (see Figure 1).

We introduce the following dimensionless branch characteristic in dimension variables:(50)$$b_{n}=\frac{\tau_{0}}{\sqrt{2}\left|\nabla \;p\right|W_{k}},\;k=\mathrm{br},\;\mathrm{out},$$

Passing to dimensionless variables and omitting the prime indexes, we obtain that
(51)bn=Re−1Fr−1bN,bN=τ˜*2|∇kp|wk,k=br,out.
where $w_{k}$ is the branch width in dimensionless variables. The parameter $b_{n}$ coincides with the Bingham number Bn for the pressure driven flows in a simple channel [46]. This is why we also call here $b_{N}$ the reduced branch Bingham number.

The reduced branch Bingham number $b_{N}$ depends on the branch width and the pressure drop along the branch. Let us consider variation of $w_{br}$ for $\alpha =90^{\circ }$, keeping $w_{out}$ constant and equal to 1. One can see from Figure 8 that a change in the side branch width implies a change in the pressure drop along the branch. Since both the branches have almost the same lengths, one can also conclude from Figure 8 that the pressure drops $|\nabla_{br}\;p|$ and $|\nabla_{out}\;p|$ in both the branches are almost equal and do not depend on the variation of the side branch width. Thus, the branch Bingham number depends mainly on the branch width.

Let us perform the results of the calculations of the time-average values $\langle b_{N}\rangle$ of the reduced Bingham number, Table 1. We remark that the pulsation interval 0<t<T is chosen in such a way that all the channel is filled with pure fluid, at t=T. Calculations reveal that T=400 if $w_{br}=w_{out}=w_{in}=1$. The terminal value *T* increases if $w_{br}$ decreases.

When we address the particles partitioning in relation to the change in $w_{br}$, as in Table 1, we arrive at the results depicted in Figure 9 and Figure 10. One can see that the concentration wave becomes slower when we reduce $w_{br}$ from 0.5 to 0.2. The total particle mass flux also reduces. As far as the polymer particles are concerned in the pulsatile injection mode, we arrive at the following conclusion: more particles fall into the branch with lower mean Bingham number. One of the implications of this result is the stoppage effect. It implies that there is no flow in the branch if the pressure gradient along the branch is small enough or the channel is very narrow. Indeed, Figure 11 shows that the solid phase velocity in the side branch is two orders of magnitude less than the solid phase velocity in the main branch in the case when the side branch is five times narrower than the main branch. Due to the regularization of Equation (Equation 7), the velocity of the solid phase in the side branch can be considered zero. The stoppage effect is well known in the Bingham fluid theory in more general context. If the pressure gradient applied in fully-developed Newtonian Poiseuille flow is suddenly set to zero, the velocity decays to zero exponentially, i.e., the theoretical stopping time is infinite [47]. This is not the case for viscoplastic or yield-stress materials [48]. The stopping time is finite.

Figure 11 and Figure 12 show the phase velocity profiles at different cross-sections of both the branches at the terminal time instant *T*. One can see that there is a correlation between the Bingham number and velocity. For $w_{br}=0.5$ and $w_{br}=0.2$, both the solid and fluid phases flow slower in the side branch than in the main branch of the width w=1. The side branch velocity decreases significantly if the width of this branch decreases. One more observation is that the velocities of both phases are almost equal due to the interphase-resistivity effect. Note, that the velocity in the main branch is indifferent to a decrease in width of the side branch. It is well known that non-Newtonian fluids in channels have a blunt Poiseuille profile [47]. The calculation results concerning the velocity field are shown in Figure 11 and Figure 12. Thus, calculated profiles are consistent with the Poiseuille-like flows and validate the numerical scheme developed in the present paper.

To show the role of the yield stress, we performed calculations with $\tau_{*}=0$, Figure 13 and Figure 14. One can see that under the conditions (Equation 43), velocities in the side branch are doubled but velocities in the main branch are unchanged.

**Remark** **1.**
*Let us comment on the validation of the numerical simulation performed in the present paper. First of all, our calculations capture the benchmark experimental Zweifach–Fung effect, stating that more particles fall into the branch where more fluid enters. A qualitative agreement with 3D experiment data on the particles partitioning is established (see Figure 7 and [43,44]) by calculations relative to the 2D branching tube. We use the same numerical algorithm as in the recent work on sedimentation [33]. The difference is only in the boundary conditions. As applied to sedimentation, our calculations also explain the Boycott effect, amounting to the fact that an inclination of a vertical vessel enhances the sedimentation of particles in suspension. It is well known that non-Newtonian fluids in channels have a blunt Poiseuille profile. Figure 11 and Figure 12 agree with such a flow property.*


In order to illustrate the convergence of the method, we performed a series of calculations for different meshes with *n* vertices on the border AB, 4n, vertices on the borders LF,NE,BL,ND, and 8n vertices on the border AC for *n* = 8, 16, 32 and 64.

This gives meshes with a total number of N= 952, 3448, 15,058 and 56,544 vertices, respectively.

We denote the numerical solution $f=(v_{s},v_{f},p,c)$ corresponding to *n* by $f^{n}$. Convergence manifests itself through the following estimates
(52)$$\underset{t_{m}}{\sup}\frac{\left|\right|f_{2n}\left(t\right)-f_{4n}\left(t\right){\left|\right|}_{L^{2}\left(\Omega \right)}}{\left|\right|f_{n}\left(t\right)-f_{2n}\left(t\right){\left|\right|}_{L^{2}\left(\Omega \right)}}\leq 2^{-r_{n}},$$
where $r_{n}=1.6,1.99\;for\;n=8,16,$ respectively. This estimate indicates that the solution is grid-independent. To guarantee convergence and an acceptable computation time, we use the mesh with at least $10^{3}$ triangular elements.

## 5. Conclusions

As in a number of studies, we consider a dense suspension of polymer particles to be a non-Newtonian fluid. Since we assume that the particles and the carrier fluid have different densities and velocities, we use a two-continuum model. The first phase of solid particles is described by the rheology of the Bingham fluid and the second phase is a viscous Newtonian fluid. There is one more rheological feature of the solid phase of particles which is associated with the generalized Fick’s law for the particle concentration flux vector. The fact is that in the full model, in addition to ordinary diffusion, barodiffusion and thermal diffusion are taken into account. In addition, the concentration flux vector also takes into account a component that depends on the gradient of the modulus of differences in phase velocities. The model agrees with the basic principles of thermodynamics and is validated through capturing the Boycott sedimentation effect. Starting from the chosen model, we address flows in a branching channel with the rather arbitrary bifurcation angle, which is reckoned from the inlet direction. We study the issue of flow partitioning and estimate the loss of particles into the side branch during the pulsatile injection of particles. We prove that the greater the bifurcation angle, the smaller the loss of particles. Our calculation of the loss agrees qualitatively with experiment data. We prove that the partitioning of particles occurs in agreement with the Zweifach–Fung effect, stating that particles prefer the branch with a higher fluid flow rate. We establish that more particles fall into the branch with a lower mean Bingham number. The results are applicable to the technology of producing microfluidic devices consisting of tubes with branches. One more application concerns the simulation of the proppant particle loss in perforations during the proppant delivery to a hydro-fracture. Now that the model has been tested on the Zweifach–Fung benchmark effect, it becomes possible to use it to study the flows of suspensions of polymer particles in confluences.

## Figures and Tables

**Figure 1 polymers-14-03880-f001:**
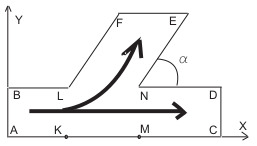
Scheme of flow domain. AB is the inlet boundary, CD and EF are the outlet boundaries. ABLK is the inlet branch, MNDC is the main branch, LFEN is the side branch. ∠END is the bifurcation angle α. The boundaries AC, BL, LF, ND, NE are impenetrable.

**Figure 2 polymers-14-03880-f002:**
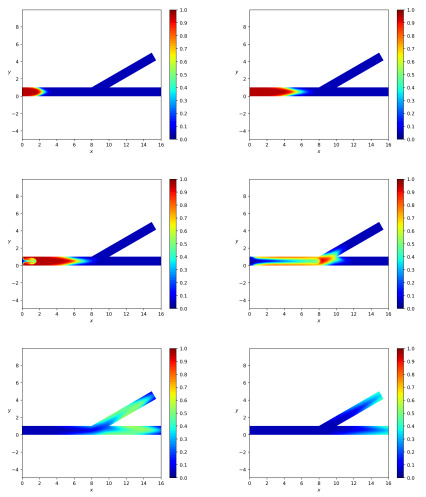
The pulsatile mode of injection. Calculated snapshots of particle mass concentration for the bifurcation angle $\alpha =30^{\circ }$ and the data (Equation 43) at the dimensionless times t=20,50,60,100,200,300 from left to right and from top down.

**Figure 3 polymers-14-03880-f003:**
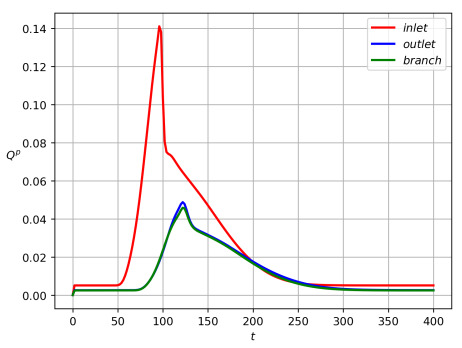
Inlet (red), outlet (blue) and side (green) particle mass flow rates $Q_{in}^{p}$, $Q_{out}^{p}$, and $Q_{br}^{p}$ versus time for the data (Equation 43) and the bifurcation angle $\alpha =30^{\circ }$.

**Figure 4 polymers-14-03880-f004:**
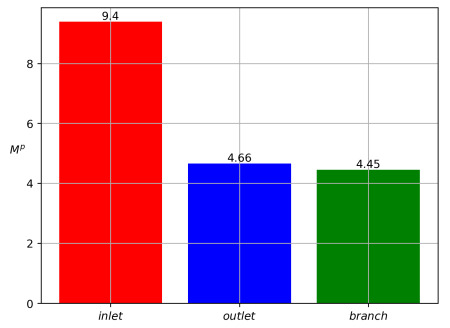
Inlet (red), outlet (blue) and side (green) values of particle masses $M_{in}^{p}$, $M_{out}^{p}$, and $M_{br}^{p}$ for the data (Equation 43) and for the bifurcation angle at $30^{\circ }$.

**Figure 5 polymers-14-03880-f005:**
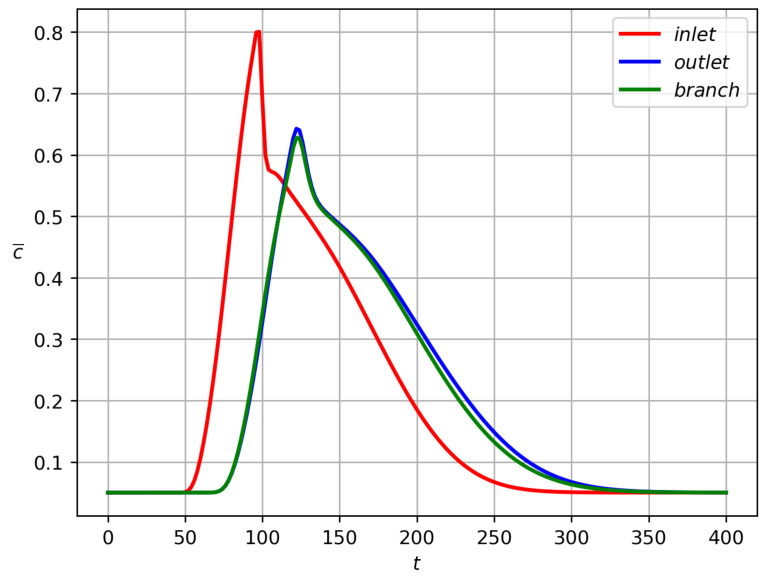
Cross-section average concentration $\bar{c}$ versus time for the bifurcation angle $30^{\circ }$ and data (Equation 43). The red, blue and green lines are for the inlet cross-section KL, for the outlet cross-section MN, and for the branch cross-section EF, respectively, (see Figure 1).

**Figure 6 polymers-14-03880-f006:**
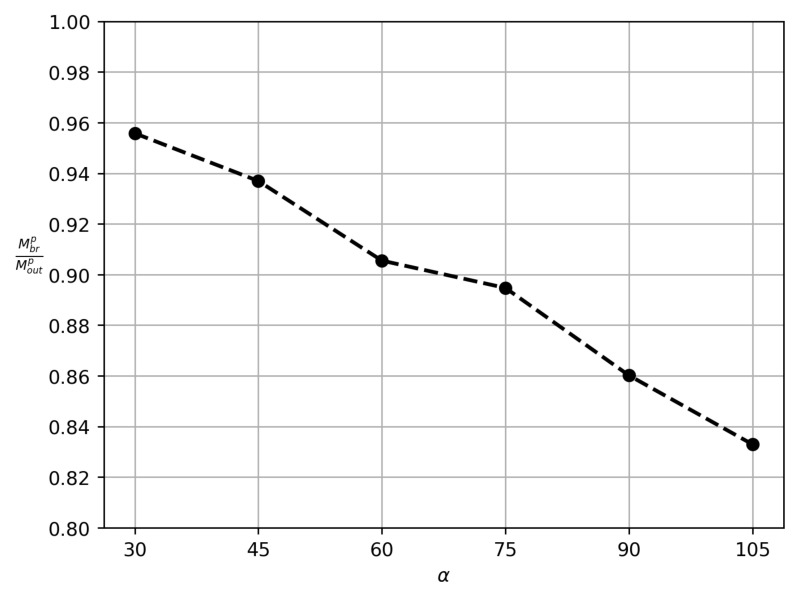
Particle-loss curve on the plane (α,y). The vertical coordinate $y=M_{br}^{p}/M_{out}^{p}$ means the relative mass loss of particles into the side branch.

**Figure 7 polymers-14-03880-f007:**
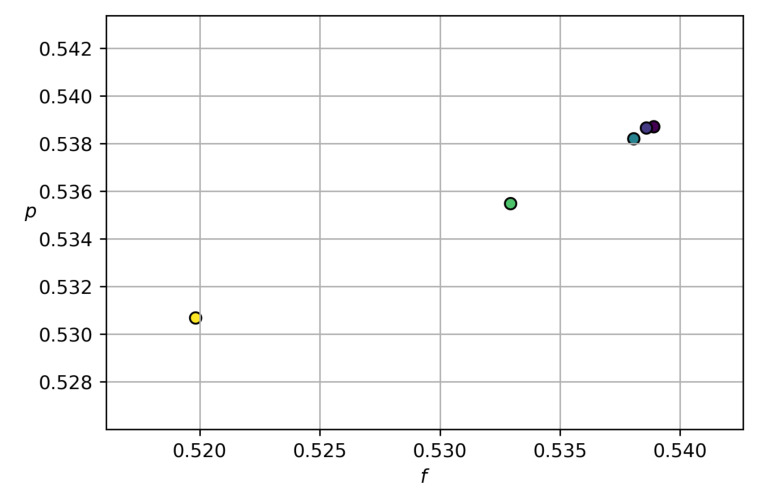
Calculated correlation between $f=M_{out}^{f}/M_{in}^{f}$ and $p=M_{out}^{p}/M_{in}^{p}$ for the bifurcation angle $\alpha =90^{\circ }$.

**Figure 8 polymers-14-03880-f008:**
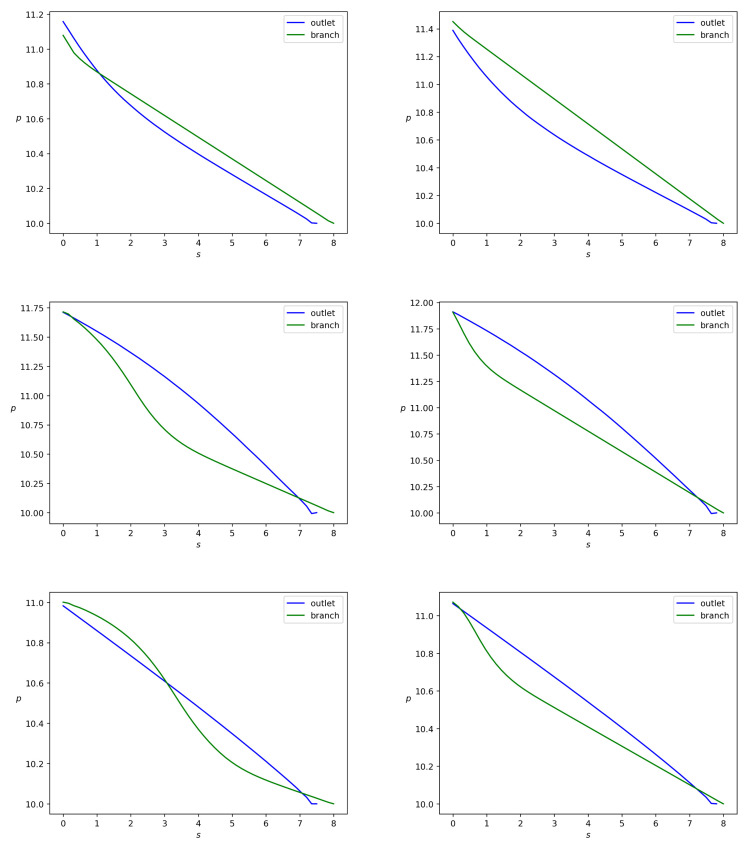
Pressure *p* versus *s*, where *s* is the distance from the bifurcation point, with $\alpha =90^{\circ }$. Green lines are for the side branch and blue lines are for the main branch. The left column corresponds to $w_{br}=0.5$ and the right column is for $w_{br}=0.2$. The pictures from top down correspond to t=100,200,300, respectively.

**Figure 9 polymers-14-03880-f009:**
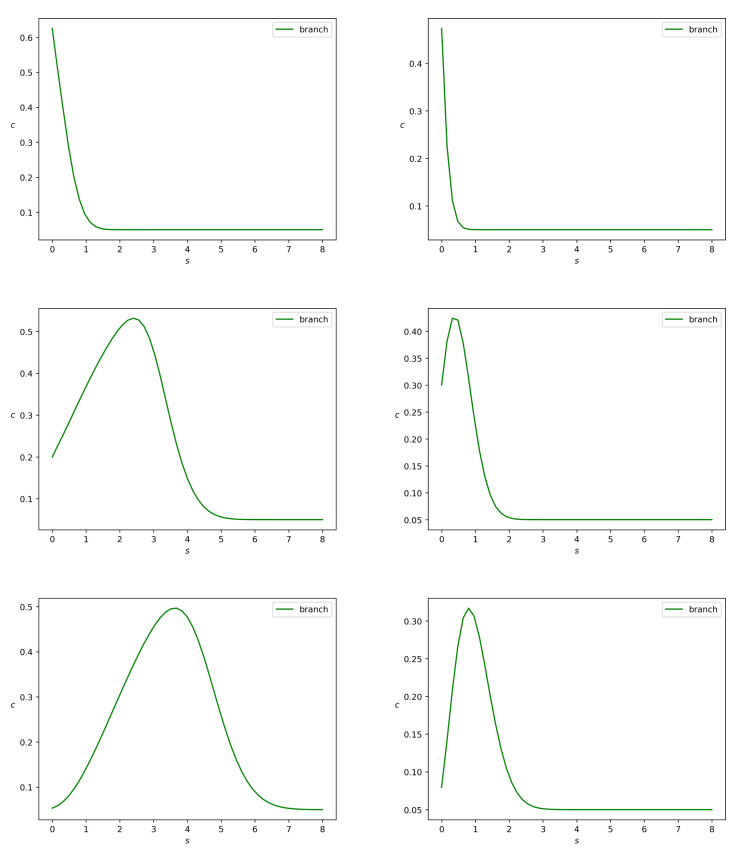
Concentration *c* versus *s* where s is the distance from the bifurcation point along the midline of the side branch with the bifurcation angle $\alpha =90^{\circ }$. The left and right pictures correspond to $w_{br}=0.5$ and $w_{br}=0.2$, respectively. The pictures from top down correspond to t=100,200,300, respectively.

**Figure 10 polymers-14-03880-f010:**
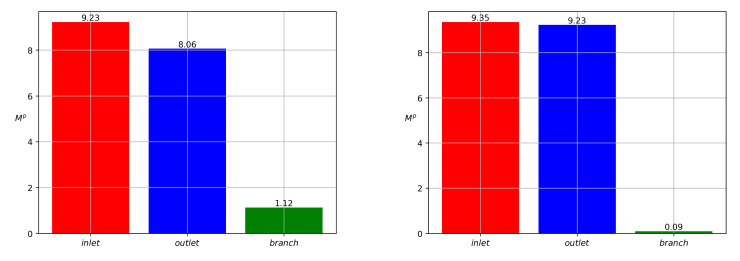
Partitioning of particles for the bifurcation angle $\alpha =90^{\circ }$. The mass of particles $M^{p}$ that passed through the branch during the period of pulsation. The left and right pictures correspond to $w_{br}=0.5$ and $w_{br}=0.2$, respectively.

**Figure 11 polymers-14-03880-f011:**
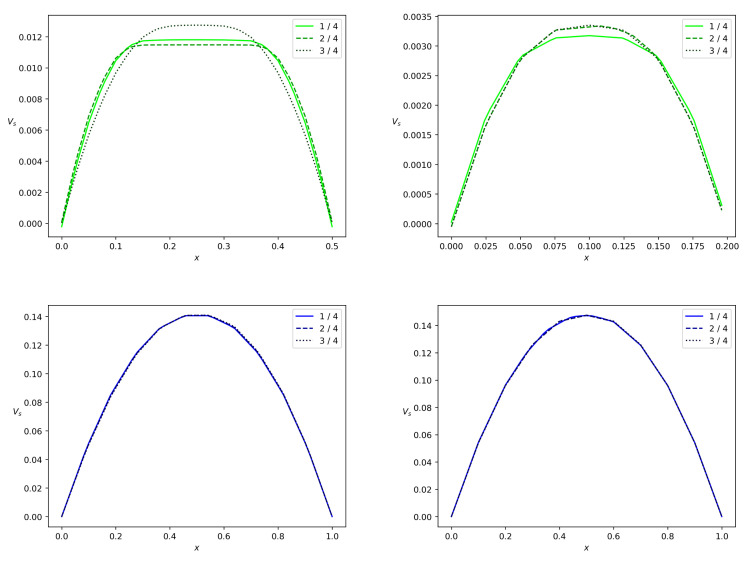
Solid phase velocity profiles in a branch for the bifurcation angle $\alpha =90^{\circ }$ at the terminal time instant. Projection $v_{s}$ of the solid phase velocity on the midline of the branch versus *x*, where *x* is a transversal variable in the branch. The side branch and the main branch are from top down, the values $w_{br}=0.5$ and $w_{br}=0.2$ are from left to right. The solid, dashed and dotted lines correspond to the branch locations 0.25l,0.5l and 0.75l reckoned from the bifurcation point.

**Figure 12 polymers-14-03880-f012:**
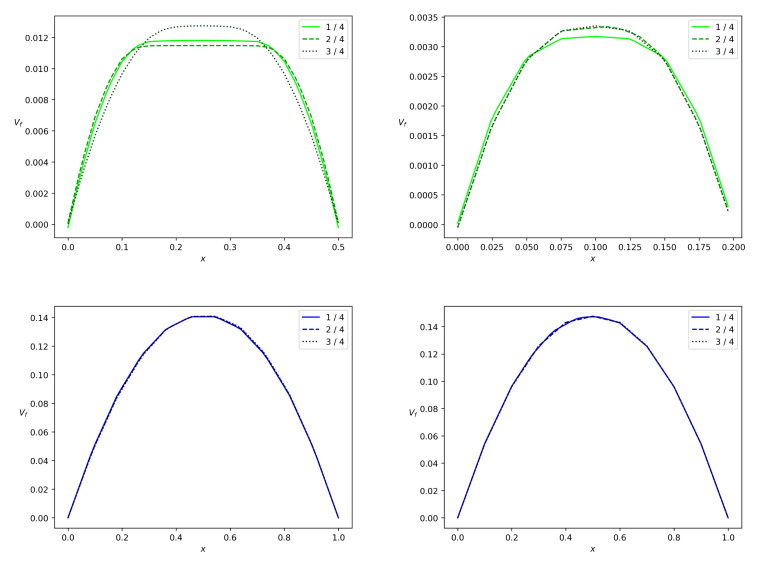
Fluid velocity profiles in a branch for the bifurcation angle $\alpha =90^{\circ }$ at the terminal time instant. Projection $v_{f}$ of the fluid velocity on the midline of the branch versus *x*, where *x* is a transversal variable in the branch. The side branch and the main branch are from top down, the values $w_{br}=0.5$ and $w_{br}=0.2$ are from left to right. The solid, dashed and dotted lines correspond to the branch locations 0.25l,0.5l and 0.75l reckoned from the bifurcation point.

**Figure 13 polymers-14-03880-f013:**
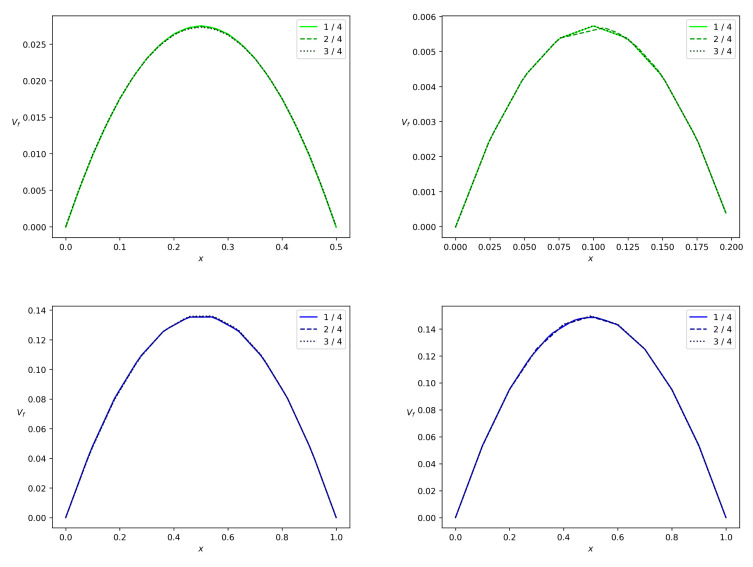
Fluid velocity profiles in a branch for the bifurcation angle $\alpha =90^{\circ }$ in the case of the zero Bingham number $b_{N}$. Projection $v_{f}$ of the fluid velocity on the midline of the branch versus *x*, where *x* is a transversal variable in the branch. The side branch and the main branch are from top down, the values $w_{br}=0.5$ and $w_{br}=0.2$ are from left to right. The solid, dashed and dotted lines correspond to the branch locations 0.25l,0.5l and 0.75l reckoned from the bifurcation point.

**Figure 14 polymers-14-03880-f014:**
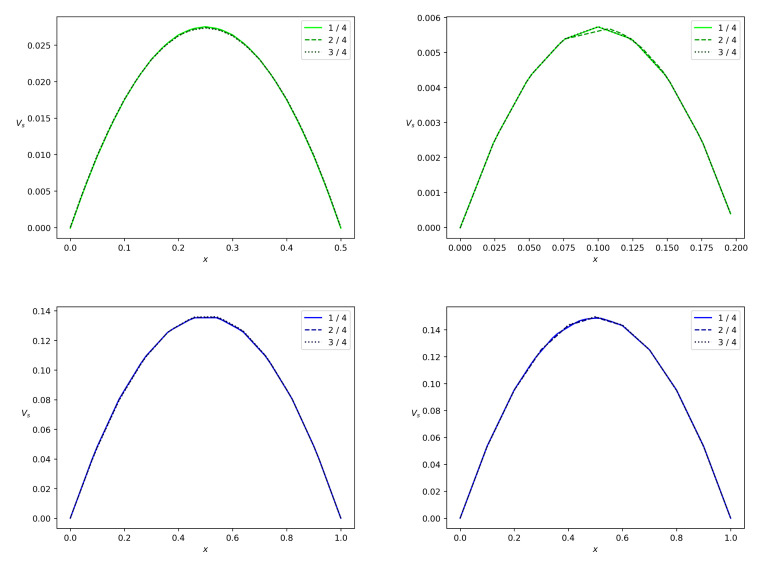
Solid phase velocity profiles in a branch for the bifurcation angle $\alpha =90^{\circ }$ in the case of the zero Bingham number $b_{N}$. Projection $v_{s}$ of the solid phase velocity on the midline of the branch versus *x*, where *x* is a transversal variable in the branch. The side branch and the main branch are from top down, the values $w_{br}=0.5$ and $w_{br}=0.2$ are from left to right. The solid, dashed and dotted lines correspond to the branch locations 0.25l,0.5l and 0.75l reckoned from the bifurcation point.

**Table 1 polymers-14-03880-t001:** Mean Bingham numbers in branches versus the width of the side branch.

$w_{br}$	1	0.5	0.2
$\langle b_{N}^{in}\rangle$	3.953	4.053	4.062
$\langle b_{N}^{out}\rangle$	7.068	4.719	4.387
$\langle b_{N}^{br}\rangle$	8.062	10.006	22.324

## Data Availability

Not applicable.

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
