# Peer review of "Flows of Dense Suspensions of Polymer Particles through Oblique Bifurcating Channels: Two Continua Approach"

_polymers, 2022, doi:10.3390/polym14183880_

Round 1

Reviewer 1 Report

Dear authors,

This manuscript presents a mathematical model proposed for dense suspension flows through channel bifurcations. The manuscript could be of interest to scientific community. I suggest that the manuscript be corrected before its possible publication.

- In general, the grammar of the manuscript can be improved before its possible publication. Some mistakes are observed along the text.

- Mention a potential application of the model, which justifies the study presented.

- In section 2, there are undefined variables.

- I think that the word “formula” can be replaced by “equation”.

- In the results section, there are ideas that must be supported with references.

- In figure 1, the X and Y axes are not observed.

- Equation 43, missing information.

- In the figures 3, 4, 5 and 6, missing information on the axes. Please add.

- The abstract and conclusions are very similar. The conclusions should be more concrete.

- According to the simulation results obtained, what is the research perspective?

Author Response

Reply to reviewer 1

Comments and Suggestions for Authors: This manuscript presents a mathematical model proposed for dense suspension flows through channel bifurcations. The manuscript could be of interest to scientific community. I suggest that the manuscript be corrected before its possible publication.

- In general, the grammar of the manuscript can be improved before its possible publication. Some mistakes are observed along the text.

Reply: We reviewed the text and corrected the errors.

- Mention a potential application of the model, which justifies the study presented.

Reply:  In Conclusions we mention: Results are applicable to the technology of producing microfluidic devices. One more application concerns the simulation of the proppant particles loss in perforations during the proppant delivery to a hydro-fracture.  

- In section 2, there are undefined variables.

Reply: After corrections, all the variable are defined.

- I think that the word “formula” can be replaced by “equation”.

Reply: We replaced “formula” by “equation” where it fits best.  

- In the results section, there are ideas that must be supported with references.

Reply:

We added 7 new references into the Results section. Ref. [39] concerns pulsatile injection, Ref. [40] concerns Poiseuille flows, Ref. [41,32] concern concentration front with cusp, Ref. [42] concerns explanation of particles partitioning with applying streamlines arguments, Ref. [32] concerns partitioning for different bifurcation angles, Ref. [46] is about the Bingham number, Ref. [47,48] explain the stoppage effect.   

- In figure 1, the X and Y axes are not observed.

Reply: The coordinate axes are labeled

- Equation 43, missing information.

Reply: The equation is corrected

- In the figures 3, 4, 5 and 6, missing information on the axes. Please add.

Reply: Names of axis are supplied.

- The abstract and conclusions are very similar. The conclusions should be more concrete.

Reply: Conclusions are corrected.

- According to the simulation results obtained, what is the research perspective?

Reply: In Conclusions we added: Now that the model has been tested on the Zweifach-Fung benchmark effect, it becomes possible to use it to study the flows of suspensions of polymer particles in confluences.

Reviewer 2 Report

The manuscript entitled Flows of dense suspensions of polymer particles through oblique bifurcating channels: two continua approach is numerical study of suspensions. I commented as follows;

1.In Fig. 3, 4, 5, 6, 8, 9, 10, 11, 12, 13, and 14, symbol in vertical axis did not exhibit. The author should revise it.

2.Vadility of the numcerical scheme is not clear. The author should revise it. Especially, the present numerical results should be compare with the experimental results.

3.The present results should be discussed after the result section.

4.The many symbols and letters are used. The author should summarize them as a nomenclature.

Author Response

Reply to Reviewer 2

Comments and Suggestions for Authors: The manuscript entitled Flows of dense suspensions of polymer particles through oblique bifurcating channels: two continua approach is numerical study of suspensions. I commented as follows;

1.In Fig. 3, 4, 5, 6, 8, 9, 10, 11, 12, 13, and 14, symbol in vertical axis did not exhibit. The author should revise it.

Reply: The coordinate axes are labeled

2.Vadility of the numcerical scheme is not clear. The author should revise it. Especially, the present numerical results should be compare with the experimental results.

Reply:

The present paper ends with a Remark: Let us comment on validation of numerical simulation performed in the present paper. First of all, our calculations capture the benchmark experimental Zweifach-Fung effect stating that more particles fall into the branch where more fluid enters. Qualitative agreement with 3D-experiment data on the particles partitioning is established (see Fig.7 and [43,44]) by calculations relative to the 2D-branching tube. We use the same numerical algorithm as in the recent work on sedimentation [33]. The difference is only in the boundary conditions. As applied to sedimentation, our calculations also explain the Boycott effect amounting to the fact that inclination of a vertical vessel enhances sedimentation of particles in suspensions. It is well known that non-Newtonian fluids in channels have a blunt Poiseuille profile. Figures 11 and 12 agree with such a flow property.

We finish the Remark by results on grid independence tests.

3.The present results should be discussed after the result section.

Reply:

We inserted four comments and a Remark into results section. 

Inset 1:

Let us discuss findings and implications concerning the results in Fig. 2. First of all, partitioning of particles occurs at the bifurcation of the channel and one can observe the propagation of the concentration wave both in the side and main branches. We emphasis that the particle mass concentration $c$ obeys the inequalities $0 < c < 1$ in spite of the fact that it is governed not by the common transport equation but satisfies a complicated diffusion equation with a matrix diffusion coefficient. Observe that we do not use any numerical cutting tricks to ensure the inclusion $0 < c < 1$. The concentration front is not blunted and there is a cusp (inverted V-shape) in the middle in agreement with experimental data [41]. For comparison, we note that the calculations in [32] do not reveal a cusp in the concentration front. The following explanation for the front with cusp can be given. The particle concentration at the center is higher compared to the near wall regions because the particles migrate from the region of high shear rate (wall) to the region of low shear rate (center). Within the framework of our model, estimates show that the transverse phase velocities are very small and the lateral migration of particles is mainly due to diffusion. Since the role of diffusion is significant, there are limitations on the scope. The results are valid for sufficiently small particles.

Inset 2:

Fig. 2 shows the qualitative partitioning of particles at the branching point of the channel with the bifurcation angle 30 degree, while Fig. 3-5 depict the quantitative partitioning. Although the side and main branches have the same width, the flow of particles into the side branch is slightly less. Apparently, this is the effect of inertia. In most papers, attempts to explain the separation of particles are associated with the study of the behavior of streamlines at the bifurcation zone [42]. Our approach is different since we consider unsteady flows. Moreover, we take into account the difference in the velocities of solid particles and fluid. In such a case the notion of streamline is not applicable.

Inset 3:

As far as the Zweifach-Fung effect is concerned, we would like not only to validate our model but to address some questions which were not captured by simulations in other publications [45]. First of all, we are interested in the pulsatile mode of injection of particles, while other authors study stationary regimes. The next question concerns particles density.  Our method is suitable for both heavy and light particles. All known calculations of flows in the branching channels concern neutrally buoyant particles. We fill this gap.

Inset 4:

One of the implications of this result is the stoppage effect. It implies that there is no flow in the branch if the pressure gradient along the branch is small enough or the channel is very narrow. Indeed, Fig.12 shows that the solid phase velocity in the side branch is two orders of magnitude less than the solid phase velocity in the main branch in the case when the side branch is five times narrower than the main branch. Due to regularization equation (7), the velocity of the solid phase in the side branch can be considered zero. The stoppage effect is well known in the Bigham fluid theory in more general context. If the pressure gradient applied in fully-developed Newtonian Poiseuille flow is suddenly set to zero, the velocity decays to zero exponentially, i. e. the theoretical stopping time is infinite [47].  This is not the case for viscoplastic or yield-stress materials [48].  The stopping time is finite.

Inset 5:

It is well known that non-Newtonian fluids in channels have a blunt Poiseuille profile [47]. The calculation results concerning the velocity field are shown in Figures  11 and  12. Thus, calculated profiles are consistent with the Poiseuille-like flows and validate the numerical scheme developed in the present paper.

4.The many symbols and letters are used. The author should summarize them as a nomenclature.

Reply:

A nomenclature is added.

Round 2

Reviewer 2 Report

The revisions are satisfied.